# Nicotinamide Riboside and Dihydronicotinic Acid Riboside Synergistically Increase Intracellular NAD^+^ by Generating Dihydronicotinamide Riboside

**DOI:** 10.3390/nu14132752

**Published:** 2022-07-01

**Authors:** Eleonora Ciarlo, Magali Joffraud, Faisal Hayat, Maria Pilar Giner, Judith Giroud-Gerbetant, Jose Luis Sanchez-Garcia, Marie Rumpler, Sofia Moco, Marie E. Migaud, Carles Cantó

**Affiliations:** 1Nestlé Institute of Health Sciences, Nestlé Research Ltd., EPFL Innovation Park, 1015 Lausanne, Switzerland; eleonora.ciarlo@rd.nestle.com (E.C.); magali.joffraud@rd.nestle.com (M.J.); judith.giroudgerbetant@irbbarcelona.org (J.G.-G.); jose-luis.sanchezgarcia@rd.nestle.com (J.L.S.-G.); marie.rumpler@rd.nestle.com (M.R.); 2Mitchell Cancer Institute, University of South Alabama, 1660 Springhill Avenue, Mobile, AL 36693, USA; fhayat@southalabama.edu; 3Nestlé Institute of Food Safety and Analytical Sciences, Nestlé Research Ltd., EPFL Innovation Park, 1015 Lausanne, Switzerland; mariapilar.giner@rd.nestle.com (M.P.G.); s.moco@vu.nl (S.M.); 4Division of Molecular and Computational Toxicology, Department of Chemistry and Pharmaceutical Sciences, Amsterdam Institute for Molecular and Life Sciences, Vrije Universiteit Amsterdam, 1081 HZ Amsterdam, The Netherlands; 5School of Life Sciences, Ecole Polytechnique Fédérale de Lausanne, 1015 Lausanne, Switzerland

**Keywords:** NAD^+^, nicotinamide riboside, dihydronicotinamide riboside, nicotinic acid riboside, dihydronicotinic acid riboside, nicotinic acid, nicotinamide, vitamin B3

## Abstract

Through evolution, eukaryote organisms have developed the ability to use different molecules as independent precursors to generate nicotinamide adenine dinucleotide (NAD^+^), an essential molecule for life. However, whether these different precursors act in an additive or complementary manner is not truly well understood. Here, we have evaluated how combinations of different NAD^+^ precursors influence intracellular NAD^+^ levels. We identified dihydronicotinic acid riboside (NARH) as a new NAD^+^ precursor in hepatic cells. Second, we demonstrate how NARH, but not any other NAD^+^ precursor, can act synergistically with nicotinamide riboside (NR) to increase NAD^+^ levels in cultured cells and in mice. Finally, we demonstrate that the large increase in NAD^+^ prompted by the combination of these two precursors is due to their chemical interaction and conversion to dihydronicotinamide riboside (NRH). Altogether, this work demonstrates for the first time that NARH can act as a NAD^+^ precursor in mammalian cells and how different NAD^+^ precursors can interact and influence each other when co-administered.

## 1. Introduction

A decrease in intracellular NAD^+^ levels has been proposed as a major hallmark for multiple age-associated health complications [1]. Supporting a causal role for NAD^+^ deficits in physiological decline, the supplementation with NAD^+^ precursors has been shown to enhance lifespan in yeast and worms [2,3,4]. While the ability of NAD^+^ precursors to enhance mouse lifespan is a matter of debate [5,6,7], multiple studies suggest that NAD^+^ precursors can prevent and treat metabolic and age-related diseases, such as fatty liver disease, kidney injury, hearing loss and neurodegenerative diseases [8]. 

The reason why NAD^+^ levels have such a pivotal role in cellular function is related to its many roles in cellular physiology. First, NAD^+^ serves as a mandatory redox co-factor for multiple metabolic reactions in the cell [9]. Second, NAD^+^ is broken down by the activities of different enzyme families that use NAD^+^ as a degradation substrate for their main catalytic activities. They include the sirtuin family of protein deacylases, the poly (ADP-ribose) polymerase enzymes (PARPs) and cyclic ADP ribose (cADPr) hydrolases [9]. By affecting the activity of these enzymes, NAD^+^ levels impact a constellation of cellular functions, including metabolic regulation, DNA repair or Ca^2+^ signaling.

Two main strategies have been devised to increase intracellular NAD^+^ and ensure its bioavailability in disease situations. One is the prevention of NAD^+^ breakdown by PARP, or cADPr hydrolase enzymes, which aims to funnel NAD^+^ towards sirtuin enzymes. Mouse models with genetic deletions of the PARP-1 enzyme or CD38 are characterized by increased NAD^+^ content in multiple tissues and higher basal SIRT1 activity [10,11]. This, in turn, provided protection against diet-induced metabolic damage [10,12]. However, PARP-1 and CD38 enzymes have critical roles in DNA repair or immune cell function, which casts doubts on the validity of this strategy for clinical use in at risk populations. An alternative approach consists in boosting NAD^+^ synthesis through the administration of NAD^+^ precursors. This strategy has gained momentum over the last decade, due to the numerous successes in pre-clinical interventions tackling metabolic and age-related diseases, as well as genetic diseases characterized by impaired DNA repair or mitochondrial dysfunction [1,8,9].

All eukaryote organisms have developed the possibility to use multiple molecules as NAD^+^ precursors [13]. In mammals, at least five independent paths exist for NAD^+^ synthesis. First, tryptophan can be used to generate NAD^+^ through the 10-step *de novo* pathway. However, tryptophan is generally considered a weak NAD^+^ precursor in humans, as the molecule is largely used for other biosynthetic purposes [14]. Nevertheless, pharmacological strategies aimed to divert tryptophan metabolism into NAD^+^ synthesis have shown success in increasing NAD^+^ availability in mouse liver and kidney [15].

A second avenue for NAD^+^ synthesis is constituted by the Preiss-Handler pathway, which transforms nicotinic acid (NA) into nicotinic acid mononucleotide (NAMN) via the nicotinic acid phosphoribosyltransferase (NAPRT), to be later converted to nicotinic acid adenine dinucleotide (NAAD) by nicotinamide mononucleotide adenylyltransferase enzymes (NMNATs) and ultimately to NAD^+^ through NAD synthase activity (NADS) [14] (Figure 1A). NA was historically used in clinical settings for the treatment of dyslipidemia and hypercholesterolemia. However, whether the therapeutic benefits of NA are related to its influence on NAD^+^ metabolism is still unclear [9]. A major limitation for the clinical use of NA is the fact that NA can activate the GPR109A receptor, which leads to some undesirable secondary effects, including a spontaneous and painful flushing [16].

A third path for NAD^+^ synthesis is based on nicotinamide (NAM) salvaging using the nicotinamide phosphoribosyltransferase (NAMPT) enzyme [14]. NAMPT catalyzes the formation of nicotinamide mononucleotide (NMN) from NAM, which is then used by NMNAT enzymes to render NAD^+^. This pathway allows cells to recycle NAM produced by NAD^+^ consuming enzymes. In this sense, mouse models of NAMPT deficiency suggest that recycling NAM is the main path by which most mammalian tissues sustain baseline NAD^+^ levels [17,18,19,20].

NAD^+^ salvage from nicotinamide riboside (NR) constitutes a fourth pathway to NAD^+^ synthesis [21]. The direct utilization of NR as a NAD^+^ precursor requires its phosphorylation by nicotinamide riboside kinase (NRK) enzymes. This generates NMN, converging then with the classic NAM salvage pathway [21,22]. Of note, a deamidated form of NR, nicotinic acid riboside (NAR), can also be phosphorylated by NRKs to generate NAMN and feed the Preiss-Handler path [23]. Finally, it was recently described that dihydronicotinamide riboside (NRH), a reduced form of NR, constitutes a new, very potent, fifth pathway towards NAD^+^ synthesis [24,25]. NRH is phosphorylated by adenosine kinase (ADK) and transformed into NMNH, then converted to NADH via NMNATs and, finally, oxidized to NAD^+^ [24] (Figure 1A).

The last decade has seen a rising interest in NAD^+^ boosting therapies for the prevention and treatment of metabolic and age-related diseases. Many of these NAD^+^ precursors are present in our diets or readily available as nutritional supplements. Further, different NAD^+^ precursors might occur simultaneously in circulation or intracellularly. However, their combined actions are poorly understood. Here, we aimed here to explore the potential interactions of different NAD^+^ precursors when applied in combination and their impact on intracellular NAD^+^ content.

## 2. Materials and Methods

### 2.1. Reagents and Chemicals

Unless otherwise specified, all reagents were purchased from Sigma-Aldrich (Merck KGaA, Darmstadt, Germany) NR chloride was kindly provided by Chromadex Inc. (Irvine, CA, USA).

### 2.2. Animal Experiments

All animal experiments were performed according to national Swiss and EU ethical guidelines and approved by the local animal experimentation committee under license VD 2770-1. All mice were in pure C57Bl/6NTac background. Mice were kept in a temperature- and humidity-controlled environment with a 12:12-h light/dark cycle. Mice had access to nesting materials and *ad libitum* access to water and a commercial low fat diet (D12450J, Research Diets Inc. (New Brunswick, NJ, USA)). Male and female mice were used indistinguishably for all experiments.

For experiments involving intraperitoneal (IP) injections, mice were fasted for 2 h and then intraperitoneally injected with either saline or different NAD^+^ precursors at the concentrations indicated in the figure legends. Then, 1 h later, mice were anesthetized using isoflurane, blood was collected through intracardiac puncture and tissues were snap frozen in liquid nitrogen. Blood samples were immediately transferred to EDTA blood collection tubes, which were kept on ice. Then, samples were centrifuged at 4 °C, 3000 rpm for 10 min in a temperature controlled tabletop centrifuge in order to isolate plasma.

### 2.3. Cell Culture

Unless otherwise specified, all common cell culture media and reagents were obtained from Gibco (Thermo Fisher Scientific Inc. (Waltham, MA, USA)). AML12 cells were cultured in Dulbecco’s modified Eagle’s medium/Nutrient mixture F-12 (DMEM:F-12), supplemented with 10% fetal bovine serum (FBS), 1× Insulin/Transferrin/Selenium solution from Gibco (ITS-G) and 100 nM dexamethasone. To evaluate the ability of compounds to stimulate NAD^+^ synthesis, cells were treated with phosphate buffer saline (PBS; as vehicle), or the different NAD^+^ precursors at the concentrations indicated.

INS1E cells were seeded in 6-well plates at a concentration of 0.8 million cells per well and then left for 24 h in growth media (RPMI 1640 complemented with 10% FBS, 1% penicillin/streptomycin (P/S), 10 mM HEPES, 1 mM sodium pyruvate, and 0.05 mM β-mercaptoethanol). MEFs were grown in DMEM medium containing 10% FBS.

All cells used in this study were verified as mycoplasma free using MycoProbe Mycoplasma Detection Kit (R%D cat. CUL001B). The chemicals inhibitors used for cell culture were the following: (E)-N-[4-(1-benzoylpiperidin-4-yl) butyl]-3-(pyridin-3-yl) acrylamide (FK866; 2 μM; from Sigma, Ref: F8557), 5′-Iodotubercidin (5-IT; 1 μM; from Tocris Bioscience, Bristol, UK, Ref: 1745), Gallotannin (100 μM; from Sigma, Ref: 403040). DMSO (1:1000) was used for the respective control groups. The concentrations are based on previous publications [24] or the manufacturer’s instructions, and were used 1 h prior to treatment with NAD^+^ precursors or PBS, as vehicle.

### 2.4. mRNA and Protein Analyses

Total mRNA extraction and cDNA conversion were performed as previously described [26]. Gene expression levels were analyzed using SYBR Green real-time PCR (Roche). The following primers were used: NAPRT: F: TGCTCACCGACCTCTATCAGG; R: CGAAGGAGCCTCCGAAAGG. β2-microglobulin: F: ATGGGAAGCCGAACATACTG; R: CAGTCTCAGTGGGGGTGAAT. Cyclophillin: F: CAGGGGAGATGGCACAGGAG; R: CGGCTGTCTGTCTTGGTGCTCTCC. Relative gene expression between genotypes was assessed through the ΔΔCt method, using β2-microglobulin and cyclophillin as housekeeping genes.

For protein analyses, cells were initially washed twice with cold PBS and lysed in lysis buffer (50 mM Tris-HCL pH 7.5, 150 mM NaCl, EDTA 5 mM, NP40 1%, sodium butyrate 1 mM, protease inhibitors), followed by centrifugation at 13,000× *g* for 10 min at 4 °C. Protein extracts from mouse tissues were isolated as previously described [27]. Cleared protein lysates were quantified using BCA assay (Pierce). For western blotting, proteins were separated by SDS-PAGE and transferred onto nitrocellulose membranes. The membranes were blocked with 5% BSA prepared in TBS-Tween 20 (TBS-T) and incubated overnight with antibodies against target proteins (NAPRT polyclonal antibody was purchased ThermoFisher Scientific, Ref# PA5-97994; GAPDH polyclonal antibody was purchased from Cell Signaling, Ref# 2118; FLAG-M2 monoclonal antibody was purchased from Sigma Ref# F1804). Anti-rabbit and anti-mouse secondary antibodies coupled to horseradish peroxidase (HRP) were purchased from Jackson (Ref# 711-035-152 and 715-035-150, respectively). Membranes were then developed by enhanced chemiluminescence (Amersham (Sigma Aldrich, Merck KGaA, Darmstadt, Germany)). For quantification, the intensity of each band was determined by densitometry using ImageJ software.

### 2.5. NAD^+^ Metabolites Measurements

NAD^+^ was extracted from both tissue and cell cultures and quantified using EnzyChrom NAD/NADH Assay kit (BioAssay Systems (Hayward, CA, USA)) according to the manufacturer’s instructions.

For NAD^+^ metabolomic analyses, biological samples (AML12 cells, 500 k–1000 k cells; plasma, 60 μL) were extracted in 1300 µL of a cold mixture of methanol:water:chloroform in 5:3:5 (*v/v*) with 5 µM NAM-D4 and 60 µL of [U]-13C-NAD^+^ labelled biomass from home-made yeast as internal standards, while keeping the samples cold throughout the procedure, following established methodologies [28]. In brief, dried samples were reconstituted in either 40 µL or 60 µL (plasma) in 60% (*v/v*) acetonitrile/water, and the supernatants were transferred into glass vials for hydrophilic interaction ultra-high performance liquid chromatography-mass spectrometry (UHPLC-MS) analysis. Positive ion mode extracted chromatograms using the MRM trace of NR, NARH, NRH, NAM, NAD^+^ and NADH were integrated and used for relative comparison, normalized to internal standard. Retention time and mass detection of metabolites were confirmed by authentic standards. All samples were randomized before analysis in experimental batches.

### 2.6. NMR

NRH, NAR and NARH were custom synthesized as described previously [29]. Twelve individual solutions of NR, NAR, NRH, and NARH were prepared in water, DMEM, and blood plasma. The concentration of NR, NAR, and NRH stock solutions in water and DMEM was 55 mM, while the concentration of the NARH stock solution in water and DMEM was 30 mM. In plasma, the stock solution was ten times diluted (5.5 and 3.0 mM, respectively) as compared to water and DMEM media. A 450 mL volume of solution was mixed with 50 μL of D2O, and the resulting mixture was vortexed three times. NMR spectral acquisition (ns = 8) was then performed in water suppression mode using a Bruker Avance III HD NMR spectrometer equipped with a 400 MHz magnet Ultrashield Plus, with temperature fixed to 300 K for all NMR measurements. TopSpin 3.2 (Bruker BioSpin (Billerica, MA, USA)) was used for all NMR spectral acquisition and preprocessing, and the automation of sample submission was performed using ICON-NMR (Bruker BioSpin). All samples were automatically shimmed. The FID was processed automatically using ICON-NMR (Bruker BioSpin), and phasing was refined manually. The solutions were examined at 1, 12, and 24 h by ^1^H NMR analysis. Chemical shift allocations for the individual compounds are shown in Appendix A.

### 2.7. Statistical Analyses

Statistical analyses were performed with GraphPad Prism version 7.02 for Windows (La Jolla, CA, USA). Differences between two groups were analyzed using a Student’s two-tailed *t*-test. Two-way ANOVA analysis was used when comparing more groups, applying Tukey’s post-hoc test. Group variances were similar in all cases. Data are expressed as mean ± S.E.M. The data that support the findings of this study are available from the corresponding authors upon reasonable request.

## 3. Results

### 3.1. The Effect of Combining Classic NAD^+^ Precursors on Cellular NAD^+^ Levels

To evaluate how different NAD^+^ precursors would interact with each other in increasing NAD^+^ levels, we made pair combinations of different NAD^+^ precursors, including NAM, NA, NR and NRH, all of which use independent paths for NAD^+^ synthesis (Figure 1A). We subsequently treated AML12 hepatocytes for 2 h with different combinations of NAD^+^ precursors and NAD^+^ levels were evaluated. 

We first explored how feeding the NAMPT path with NAM interacts with other pathways. NAM acts as a very weak extracellular NAD^+^ precursor in AML12 cells, only leading to significant increases in NAD^+^ levels when using millimolar concentrations (Appendix A). Therefore, we used NAM at a 5 mM concentration in all the experiments from here on. Even at this very high concentration, the effects were modest compared to those of NR or NRH at 0.5 mM and 0.01 mM, respectively (Appendix A and Figure 1B). These concentrations of NR and NRH were chosen to achieve comparable, submaximal, effects on NAD^+^ levels. The combination of NAM and NR did not render any significant difference compared to the single NR treatment (Figure 1B), suggesting that the NAMPT and NRK1 paths do not act in an additive fashion on NAD^+^ levels. A similar outcome was obtained with NRH, where no additivity with NAM was observed (Figure 1B).

We also analyzed how the NAMPT and Preiss-Handler paths interacted. However, NA failed to increase NAD^+^ levels in AML12 hepatocytes, even at a concentration of 5 mM (Figure 1B and Appendix A). This result echoes previous observations in HepG2 cells [29] and suggests that AML12 might harbor poor NAPRT activity. To confirm this point, we first analyzed the mRNA levels of NAPRT in AML12 hepatocytes compared to mouse tissues. The results highlighted that NAPRT expression is detected in liver and kidney tissue, yet barely detectable in muscle and AML12 cells (Appendix A). We next evaluated NAPRT protein levels. As in HepG2 cells or muscle tissue, NAPRT protein levels were close to undetectable in AML12 cells (Appendix A). To avoid any confusion with other bands detected by the NAPRT antibody, we transfected AML12 and HepG2 hepatocytes with FLAG-NAPRT, which further confirmed that NAPRT is not expressed in AML12 or HepG2 hepatocytes (Appendix A). In HepG2 cells, overexpression of NAPRT is enough to allow NA-induced NAD^+^ synthesis [29] (Appendix A). However, this was not the case in AML12 (Appendix A). To test the interaction between NA and other NAD^+^ precursors, we used HepG2 cells overexpressing mouse NAPRT. The results indicate that, as with NAM, the actions of NA and other precursors are not additive (Figure 1C). Finally, the combination of NR and NRH also failed to render additive effects (Figure 1D). Altogether, these experiments suggest that NRH, NR, NAM and NA do not additively increase NAD^+^ content under regular cell culture conditions in hepatocytes, indicating that their actions are counter-regulated.

### 3.2. Exploring the Actions of NAR and NARH

Like NR, NAR can also be used as a NAD^+^ precursor via phosphorylation by NRK1, which transforms it into NAMN, then feeding the classic Preiss-Handler path (Figure 2A). The effectiveness of NAR to act as a NAD^+^ precursor in mammalian cells and organisms has been poorly explored. Therefore, we performed dose-response experiments in cultured hepatocytes, showing that NAR effectively increases NAD^+^ levels in AML12 cells, reaching significant elevations at 0.05 mM concentrations (Figure 2B). 

We have recently reported how NRH, a reduced form of NR, had unique biological properties compared to its oxidized counterpart, NR [24]. We hence wondered if the same would occur with a reduced form of NAR, dihydronicotinic acid riboside (NARH) (Figure 2A). NARH dose-dependently increased NAD^+^ levels in AML12 hepatocytes, reaching significant differences at 0.1 mM concentrations (Figure 2C). The amplitude of the effect of NARH was modest, and comparable to those of its oxidized counterpart, NAR (Figure 2C). In vivo, NAR and NARH increased hepatic NAD^+^ after intraperitoneal administration in mice (Figure 2D). Under the same circumstances, NARH failed to significantly increase NAD^+^ levels in muscle and kidney, while NAR displayed very mild effects compared to those of NR and NRH (Figure 2D).

When NAR was combined with NAM, NR or NRH, NAD^+^ levels were always similar to those obtained with the strongest precursor as a single agent (Figure 3A). In contrast, NARH achieved synergistic effects with NR in increasing NAD^+^ levels, but not with any of the other NAD^+^ precursors tested (Figure 3B). This did not only occur in hepatocytes, as the dramatic boost in NAD^+^ content after the simultaneous exposure to NR and NARH could also be observed in INS1E pancreatic β-cells and in mouse embryonic fibroblasts (MEFs) (Figure 3C). The effect was remarkable, given the fact that NR or NARH had limited effects on NAD^+^ levels in these two cell lines when used as single treatments (Figure 3C). The synergistic action of NR and NARH could be confirmed in vivo. When NR and NARH were injected intraperitoneally in mice we could observe that the co-treatment showed a much more potent effect on hepatic and renal NAD^+^ levels than any of the single treatments (Figure 3D). Interestingly, the synergistic effects in the liver were observed irrespectively of whether NR and NARH were administered combined in a single mixture or when each compound was administered separately in different parts of the body (Figure 3D). This synergism between NR and NARH was less marked in kidney than in liver, probably because NR has already very strong effects in kidney NAD^+^ levels due to the very high expression of NRK1 [22].

### 3.3. Understanding NAD^+^ Synthesis upon NR and NARH Administration

The above results drove further exploration of the metabolic pathways by which the combination of NR and NARH can affect NAD^+^ synthesis. The results in Figure 3B indicate that NR degradation to NAM was not a critical step, as NAM and NARH did not lead to cooperative increases in NAD^+^ levels. Accordingly, inhibition of NAMPT with FK866 did not alter the ability of the NR + NARH combination to increase NAD^+^ levels (Figure 4A). The combination was effective in AML12 cells, lacking NAPRT, indicating that the effect does not rely on the generation of NA.

The vigorous increase in NAD^+^ levels observed when combining NR and NARH was reminiscent of the effects of NRH. Therefore, we next tested if ADK could be involved in the actions of the NR and NARH combination. While ADK inhibition did not affect the action of NR or NARH when used separately, it totally abrogated their synergistic action (Figure 4B). The second critical step for NRH-induced NAD^+^ synthesis is mediated by NMNAT enzymes [24]. Their inhibition with gallotannin also largely prevented NAD^+^ synthesis by NR and NARH, either alone or in combination (Figure 4C).

### 3.4. NR and NARH Lead to NRH Synthesis

The above results suggest that the combination of NR and NARH requires a path like that of NRH to increase NAD^+^ levels. The simplest explanation for this would be that the combination of NR and NARH led to the production of NRH. To evaluate this possibility, we analyzed the intracellular levels of NRH in AML12 cells after treatment with either NR, NARH or both for 2 h. The results showed that, when used together, NR and NARH increased the intracellular levels of NRH (Figure 5A).

To investigate if this production of NRH occurred extracellularly or intracellularly, and whether it required an enzymatic activity, we initially evaluated whether NR and NARH led to NRH formation when combined in water. Simply adding the two molecules to a vial of water at room temperature was enough to generate NRH (Appendix A). Similar observations were made when both compounds were spiked in mouse serum (Appendix A). Accordingly, the intraperitoneal administration of NR and NARH led to increased blood levels of NRH, either when both compounds were administered together or separately (Figure 5B). The effects of NR and NARH when applied separately were lower than when applied together, probably because the compounds might be partially diluted in circulation or metabolized by tissues before they interact with each other. In line with previous results [22], we could not detect NR in circulation 1 h after a NR intraperitoneal injection (Figure 5B). Interestingly, NR could be detected after co-administration of NR and NARH (Figure 5B), in line with the observation that NRH administration leads to detectable levels of circulating NR [24]. Altogether, these observations suggest that NR and NARH can produce NRH extracellularly in the absence of an enzymatic activity.

We next performed ^1^H NMR measurements in water, DMEM, and mice plasma spiked with 10% (*v/v*) D_2_O to demonstrate that NRH is formed by a transhydrogenation reaction between NR and NARH. A two-fold excess NR was used in these experiments to ensure that the new resonances appearing in the NMR spectrum belonged to product formation and were not artefacts of protein binding or pH effects. As the spectra display (Figure 6), NR and NARH immediately led to the formation of NRH and NAR upon mixing. This is observed by the loss of the NARH signals at 6.88 ppm and the appearance of the NAR signal at 8.34 ppm, with the concomitant disappearance of the NR signal at 9.50 ppm and the appearance of the NRH signal at 7.03 ppm. The transhydrogenation occurs at similar rate whether the reaction takes place in water, DMEM or plasma (Figure 6 and Appendix A). However, this is not a bidirectional reaction, as mixing NRH and NAR did not lead to the formation of NR and NARH (Figure 6) [30].

In order to rule out that the intracellular utilization of NR and NARH could be the explanation for NRH production when cells are exposed to these compounds, we incubated cells with one compound, either NR or NARH, for 30 min, and, after a quick washout with PBS, cells were then placed with the second compound. The results indicate that the synergism on NAD^+^ synthesis only occurs when cells are incubated with NR and NARH simultaneously, but not when these two compounds are provided serially (Appendix A). Collectively, our results indicate that NARH and NR lead to synergistic effects on NAD^+^ synthesis due to their chemical interaction in the extracellular milieu.

## 4. Discussion

Most eukaryote cells are equipped with the enzymatic machinery required to use different molecules as precursors for NAD^+^ and NADH [9]. This suggests an evolutive pressure to avoid relying on a single molecule to maintain intracellular NAD^+^ levels. To date, however, we do not fully understand how the different NAD^+^ biosynthetic paths interact and if their actions can be additive.

This study indicates that NAD^+^ precursors do not additively increase NAD^+^ levels in cultured mouse hepatocytes. The net action on NAD^+^ levels of any given combination between two NAD^+^ precursors equaled that of the most potent compound of the two. One limitation of the study is that it does not evaluate NAD^+^ metabolic fluxes, so we cannot rule out that higher NAD^+^ synthesis occurs when the combination is applied, yet it is counteracted by higher consumption. A second limitation is that we measured NAD^+^ as our main readout, yet these combinations might additively influence other NAD^+^-related molecules. This is a critical point, as NAD^+^ biosynthesis paths are intracellularly interconnected. For example, the administration of NR or NRH leads to an increase in the levels of NAAD [24,31], a metabolite related to the Preiss-Handler path. It also leads to a substantial increase in NADP(H) [31]. Similarly, NRH or NAM treatment can lead to NR synthesis [24,32], while NA can lead to NAR synthesis [32]. All these observations suggest that NAD^+^ levels are tightly regulated and that an excessive flux towards NAD^+^ could be diverted into other NAD^+^-related paths.

The recent demonstration that NRH uses a new path for NAD^+^ synthesis [24,25] illustrates the fact that our knowledge of NAD^+^ metabolism is not complete. Our data characterizes an additional molecule that can act as a NAD^+^ precursor in mammalian cells and tissues, i.e., NARH. NARH is the reduced form of NAR, in the same way that NRH is for NR. NARH, however, is a weak NAD^+^ precursor, that achieves, at best, similar effects to those of NAR. The exact path used by NARH to synthesize NAD^+^ is still unresolved, although our results provide some indications. First, unlike NRH, NARH does not require ADK to induce NAD^+^ synthesis, but requires NMNATs. NARH gets easily chemically oxidized to NAR in the absence of a co-oxidant and given the striking similarity in their effects, it is likely that oxidized NARH, i.e., NAR [23], uses the NRK path to initiate its routing towards NAD^+^ synthesis (Figure 7A). To synthesize NAD^+^, NAR and NARH must be amidated. This step should likely occur via the glutamine dependent NAD synthases (NADSYN). NADSYN is expressed in the small intestine, testis, liver and kidney, but very weakly in other tissues [33]. Accordingly, NAR and NARH increased NAD^+^ levels in liver, but failed to do so in cultured fibroblast, pancreatic β-cell lines or in skeletal muscle.

We report here for the first time that AML12 hepatocytes, as HepG2 cells [34,35], lack NAPRT. The loss of expression of the NAPRT gene has been observed in multiple tumors and correlates with the hypermethylation of the CpG island that overlaps with the transcription start site of NAPRT [36]. This finding rules out the possibility that the ability of NAR and NARH to increase NAD^+^ level requires the cleavage of these molecules into NA. Interestingly, NAPRT overexpression was enough to allow HepG2, but not AML12, cells to respond to NA. One possibility is that the NAPRT overexpression levels in HepG2 were higher than those on AML12. Alternatively, our results could suggest that AML12 harbor further defects that compromise the functionality of the Preiss-Handler pathway. 

To date, the dietary presence and physiological roles of NAR and NARH remain poorly explored. Some data, however, suggests that NAR could be intracellularly synthesized. NAR was detected as an intracellular metabolite when yeast was grown in media containing NA [37,38]. Under these circumstances, NAR appeared as a product of NAMN dephosphorylation by 5‘-nucleotidases [37] or the Pho8 phosphatase [38]. Recent observations in HEK293, HepG2 and HeLa cells also show that NAR could be intracellularly synthesized from NAMN in mammalian cells [32]. That study also demonstrated that the generation of NAR in mammalian cells can be coupled to its release to the extracellular milieu, where it could act as a NAD^+^ precursor in a paracrine fashion [32].

While the natural existence of NARH as a natural intra- or extra-cellular metabolite has not yet been characterized, one could speculate multiple avenues driving its formation. A recent manuscript has described how the glucose 6-phosphatase dehydrogenase (G6PD) enzyme can promote the reduction of NAADP to NAADPH [27]. The dephosphorylation of NAADPH by intracellular phosphatase enzymes to NAADH could lead to the formation of NARH by multiple pathways. In addition to the two phosphatases already characterized for NAADP dephosphorylation, NAADPH could also be a substrate for MESH1 and NOCT and converted to NAADH, in a similar manner to NADP and NADPH being converted to NAD and NADH [39,40]. NAADH could then be subject to a sequential transformation to NAMNH by pyrophosphatases or NMNATs, and from NAMNH to NARH by phosphatases, mirroring the reversible transformation of NAAD to NAMN to NAR. Alternatively, the redox potential of NAD(H) is at −340 mV while that of NAAD(H) is at −400 mV, rendering the NAADH oxidation with concomitant reduction of NAD^+^ to NADH energetically favorable. In cells, NAADH could be oxidized to NAAD by chemical transhydrogenation in the presence of NAD(P) to generate NAD(P)H. Similarly, the rapid hydride transfer observed between NARH and NR could also happen between NAADPH and NAD(P), thus regenerating NAADP and reversing the role of G6PD catalysis, without the need for an oxidase. Although, the role of DUOX1 and DUOX2 in the oxidation of NAADPH to NAADP has been demonstrated in T cells, chemical proximity of NAADPH with NAD(P) species might facilitate the transhydrogenation observed for NARH and NR and contribute to the reactivation of NAADP overtime, unless NAADP conversion to NAAD takes place.

The physiological relevance of these reduced NAD^+^ precursors on global NAD^+^ homeostasis in health and disease will be a major focus of interest in the future. Further studies will be required to understand their therapeutic potential and clinical use. While acute NRH treatments in mice have been reported [24], no study to date has evaluated the safe dose range or potential side effects of NRH or NARH when administered chronically in vivo. Therefore, reduced NAD^+^ precursors are not yet apt for dietary supplementation or clinical application in humans. One of the frontrunner NAD^+^-boosting compounds, NR, shows limited bioavailability after oral administration [24], which can curtail its therapeutic success. We demonstrate here that, when combined with NARH, NR can be more pronouncedly detected in circulation. Therefore, one could speculate about NR+NARH co-administration as a way to enhance NR-driven NAD^+^ synthesis in tissues. Our study, however, is not without limitations. First, the tests performed are of a limited size, which is not optimal for parametric statistics. Second, the additivities and interactions of different NAD^+^ precursors have been studied in AML12 hepatocytes, and we cannot rule out that different outcomes might occur in other tissues or cell types.

## Figures and Tables

**Figure 1 nutrients-14-02752-f001:**
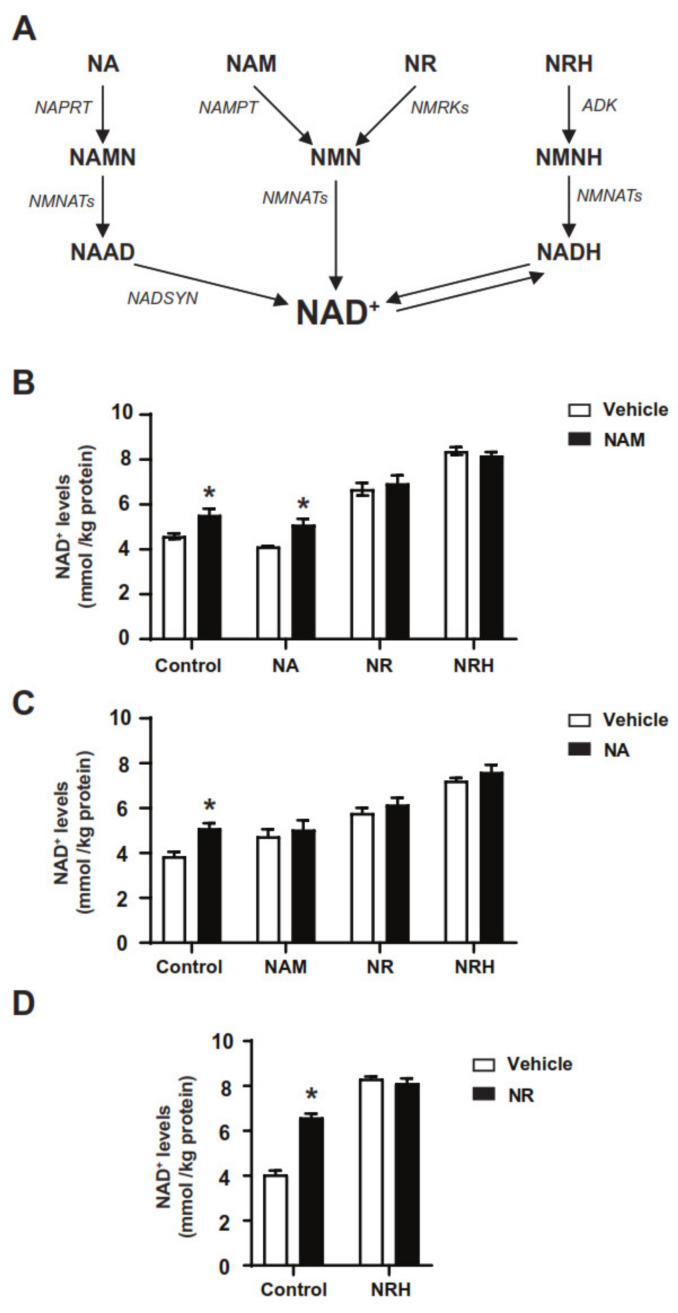
The different routes for NAD^+^ synthesis and their combinatorial effects. (**A**) Scheme depicting the five main NAD^+^ synthesis routes in mammals. NA: nicotinic acid; NAM: nicotinamide; NR: NAM riboside; NRH: dihydronicotinamide riboside; NAMN: NA mononucleotide; NMN: NAM mononucleotide; NMNH: dihydronicotinamide mononucleotide; NAAD: NA adenine dinucleotide; NAD: nicotinamide adenine dinucleotide in its reduced (NADH) or oxidized (NAD^+^) form. NAPRT: NA phosphoribosyltransferase; NAMPT: NAM phosphoribosyltransferase; NMRKs: NR kinases; ADK: adenosine kinase; NMNATs: NMN adenylyltransferases; NADSYN: NAD synthase. (**B**) AML12 were treated with either PBS (as vehicle) or different NAD^+^ precursors (NR at 0.5 mM, NRH at 0.01 mM, NAM at 5 mM, NA at 5 mM) for 1 h. Then, acidic extracts were obtained to measure NAD^+^ levels. (**C**) HepG2 cells were transfected with mouse NAPRT and, 48 h later, they were treated for 1 h with either PBS (as vehicle) or different NAD^+^ precursors at the doses indicated in (**B**). Then, 1 h later, acidic extracts were obtained to measure NAD^+^ levels. (**D**) AML12 were treated with either PBS (as vehicle), NR (0.5 mM), NRH (0.01 mM) or both for 1 h. Then, acidic extracts were obtained to measure NAD^+^ levels. All data is expressed as mean +/− SEM of *n* = 3 experiments. * Indicates *p* < 0.05 vs. the respective vehicle treated group.

**Figure 2 nutrients-14-02752-f002:**
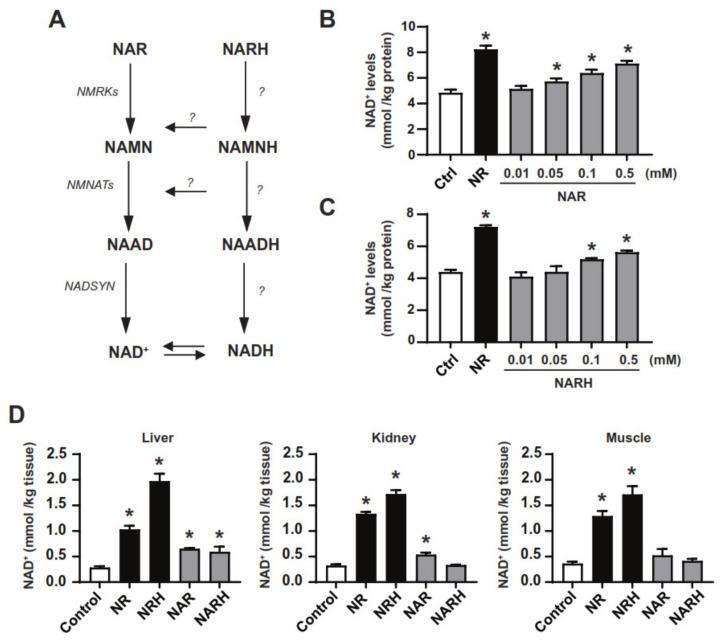
NAR and NARH as NAD^+^ precursors. (**A**) Potential NAD^+^ synthetic routes from NAR and NARH. (**B**) AML12 were treated in a dose-response fashion with PBS (as control), NR (0.5 mM) or increasing concentrations of NAR for 1 h. Then, acidic extracts were obtained to evaluate NAD^+^ levels. (**C**) AML12 were treated in a dose-response manner with PBS (as control), NR (0.5 mM) or increasing concentrations of NARH for 1 h. Then, acidic extracts were obtained to evaluate NAD^+^ levels. (**D**) Mice, *n* = 4–6 per group, were fasted for 2 h and administered intraperitoneally with saline (as control) or 500 mg/kg of NR, NRH, NAR or NARH. Then, 1 h later, liver, kidney, and quadriceps muscle were collected and flash-frozen to evaluate NAD^+^ levels. All values are presented as mean ± SEM of *n* = 6 independent experiments (**B**,**C**) or *n* = 4 animals per group (**D**). * Indicates *p* < 0.05 vs. the control group.

**Figure 3 nutrients-14-02752-f003:**
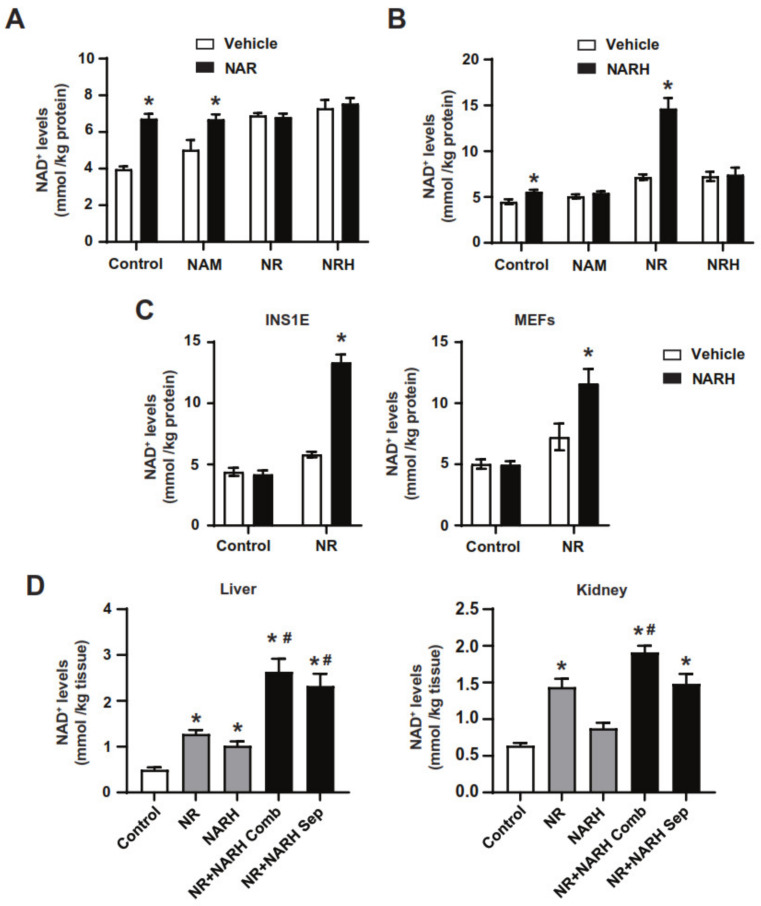
Interaction of NAR and NARH with other NAD^+^ precursors. (**A**) AML12 were treated with PBS (as vehicle) or combinations of NAD^+^ precursors as specified in the figure (NAR, 0.5 mM; NR, 0.5 mM; NAM, 5 mM; NRH, 0.01 mM). One hour later, acidic extracts were obtained to measure NAD^+^ levels. (**B**) AML12 were treated with PBS (as vehicle) or combinations of NAD^+^ precursors as specified in the figure (NARH, 0.5 mM; NR, 0.5 mM; NAM, 5 mM; NRH, 0.01 mM). One hour later, acidic extracts were obtained to measure NAD^+^ levels. (**C**) INS1E and MEF cells were treated with either PBS (as vehicle), NR (0.5 mM), NARH (0.5 mM) or NR+NARH for 1 h. Then, acidic extracts (**D**) After a 2 h fast, mice were intraperitoneally injected with saline (as control vehicle) or 500 mg/kg of NR, NARH, or NR+NARH. For the NR+NARH combination, we had one group in which compounds were injected using a single mixture with both compounds and a second group where the compounds were injected separately, one in the left side of the intraperitoneum and one in the right side. One hour later, liver tissue was snap-frozen and NAD^+^ levels were measured. All data are presented as mean +/− SEM of *n* = 4 independent experiments (**A**–**C**) or *n* = 5 mice per group (**D**). * Indicates *p* < 0.05 vs. vehicle treated group. # Indicates *p* < 0.05 vs. NR and NARH groups.

**Figure 4 nutrients-14-02752-f004:**
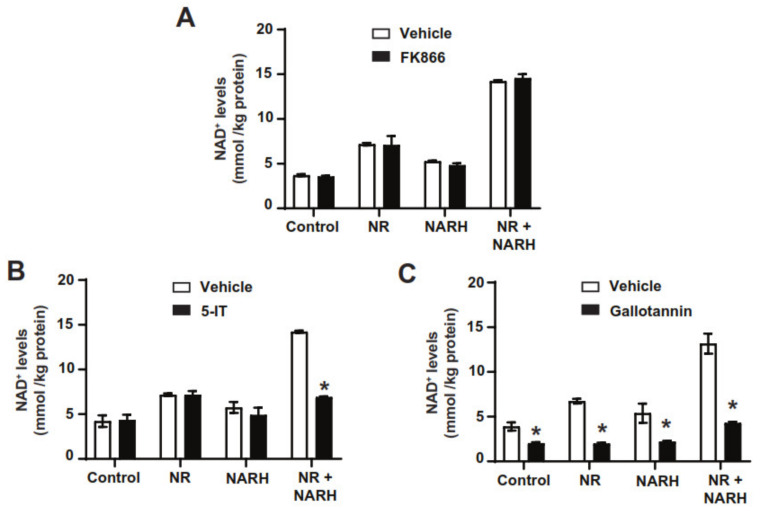
Unravelling the metabolic path by which NR + NARH increase NAD^+^ levels. (**A**) AML12 were treated with DMSO (as vehicle) or FK866 (2 µM) for 1 h and then treated with either PBS, NR (0.5 mM), NARH (0.5 mM) or both. Two hours later, acidic extracts were obtained to evaluate NAD^+^ levels. (**B**) AML12 were treated with DMSO (as vehicle) or 5-IT (1 µM;) for 1 h and then treated with either PBS, NR (0.5 mM), NARH (0.5 mM) or both. Two hours later, acidic extracts were obtained to evaluate NAD^+^ levels. (**C**) AML12 were treated with DMSO (as vehicle) or gallotannin (100 μM) for 1 h and then treated with either PBS, NR (0.5 mM), NARH (0.5 mM) or both. Two hours later, acidic extracts were obtained to evaluate NAD^+^ levels. All data are presented as mean +/− SEM of *n* = 3 independent experiments. * Indicates *p* < 0.05 vs. the respective vehicle treated group.

**Figure 5 nutrients-14-02752-f005:**
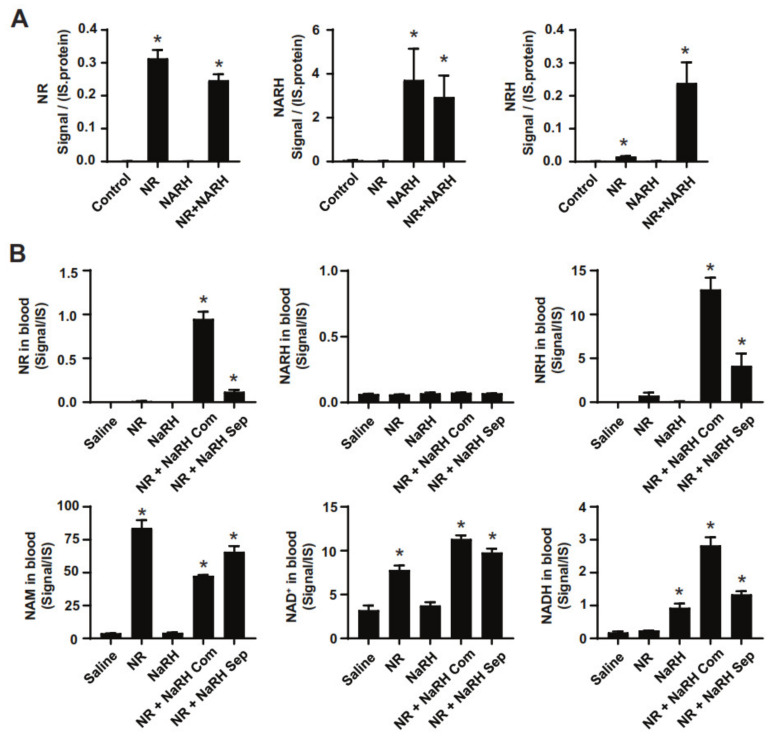
The NR + NARH combination leads to the generation of NRH. (**A**) AML12 were treated with either PBS (as control), NR (0.5 mM), NARH (0.5 mM) or both. Then, 2 h later, cells were flash-frozen and processed for metabolomic analyses. (**B**) After a 2 h fast, mice were intraperitoneally injected with saline (as vehicle) or 500 mg/kg of NR, NARH, or NR+NARH. For the NR+NARH combination, we had one group in which both compounds were injected using a single mix with both compounds and a second group where the compounds were injected separately. One hour later, blood was collected and the blood levels of different NAD^+^-related metabolites were evaluated. All values are presented as normalized LC-MS signal intensities, mean +/− SEM of *n* = 3 independent samples/group (**A**) or *n* = 5 mice per group (**B**). * indicates *p* < 0.05 vs. the control (**A**) or saline-injected (**B**) group.

**Figure 6 nutrients-14-02752-f006:**
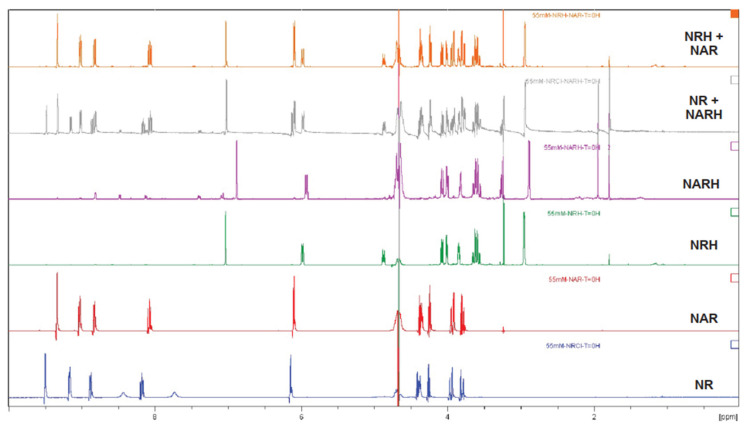
Transhydrogenation between NR and NARH. Proton NMR spectra of NR (NR chloride), NAR, NARH, NRH, NR+NARH and NRH+NAR in water (10% D_2_O).

**Figure 7 nutrients-14-02752-f007:**
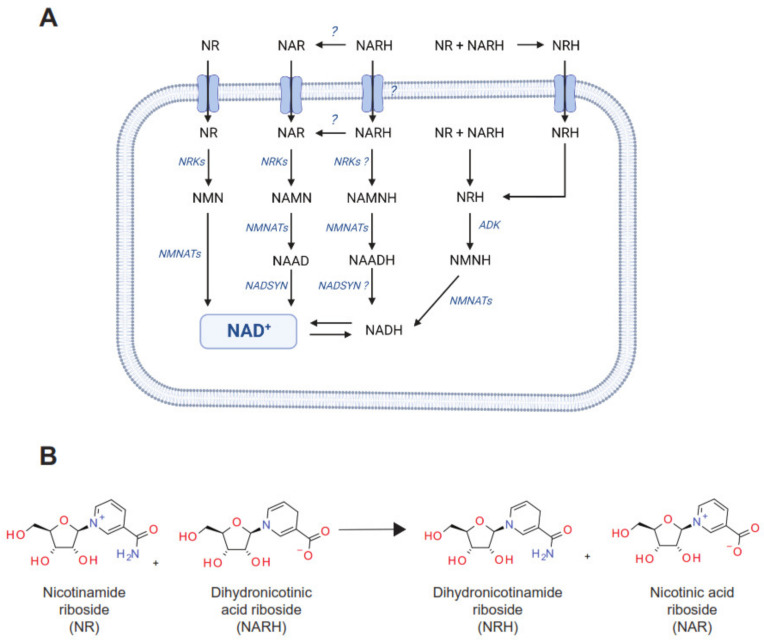
Different paths towards NAD^+^ synthesis from ribosylated NAD^+^ precursors. (**A**) Nicotinamide riboside (NR), nicotinic acid riboside (NAR) and their respective reduced forms, dihydronicotinamide riboside (NRH) and dihydronicotinic acid riboside (NARH), act as ribosylated NAD^+^ precursors. Ribosylated NAD^+^ precursors are characterized by having a ribose moiety already bound to the nicotinamide (NAM) or nicotinic acid (NA) rings (see (**B**)). After entering the cell, NR is phosphorylated by NR kinases (NRKs), generating nicotinamide mononucleotide (NMN), which is then used by the NMN adenylyltransferase enzymes (NMNATs) to generate NAD^+^. NRKs also initiate the path of NAR towards NAD^+^ synthesis by generating NA mononucleotide (NAMN). NAMN is then used to generate NA adenine dinucleotide (NAAD) through the activity of NMNATs to be later amidated to NAD^+^ by the NAD synthase (NADSYN) enzyme. In this work we describe how the reduced version of NAR, NARH can also increase NAD^+^ levels in cultured cell lines. This could occur (1) via its extra- or intracellular oxidation to NAR, or (2) via a path rendering reduced intermediates, such as dihydronicotinic acid mononucleotide (NAMNH) and dihydronicotinic acid adenine dinucleotide (NAADH). The enzymes catalyzing these reactions, however, remain unclear. This work also uncovers how the combination of NR and NARH generates NRH in a quick, non-enzymatic fashion through the reaction depicted in (**B**). This phenomenon occurs extracellularly but could also occur if these molecules encountered intracellularly. NRH leads to NAD^+^ synthesis through a path initiated by the phosphorylation of NRH by adenosine kinase (ADK), generating dihydronicotinamide mononucleotide (NMNH), which is then used by NMNATs to generate NADH, which is then oxidized to NAD^+^.

## Data Availability

Raw data are available upon reasonable request.

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
