# Peer review of "Nicotinamide Riboside and Dihydronicotinic Acid Riboside Synergistically Increase Intracellular NAD+ by Generating Dihydronicotinamide Riboside"

_nutrients, 2022, doi:10.3390/nu14132752_

Round 1

Reviewer 1 Report

The manuscript presented by Ciarlo et al. found a new molecule to increase NAD+ and a synergistic effect between two NAD+ modulators. The work is well structured and designed. Some minor issues should be fixed.

-The authors use NRH abbreviator in the abstract without define it.

-In p.3 l.126 "Cells" should be "cells"

-In section 2.5 also secondary antibodies should be listed

_It should be desiberable a graphic relating NR, NRH and NARH, if possible (similar to those in figs 1A and 2A). Moreover NRK1 is cited in the text, but is not present in any of these figures.

-In figure 3D (right panel, kidney) the effect of NR is similar to NR+NARH administered separately. In figure 5D (NRH and NADH) the is also different if the compounds are administered separately. It is important to comment and discuss this results.

-The major limitation of the study is that, although these compounds are proposed as suplements, any toxicity assay is done . This limitation should be mentioned and discussed. If there are any previous toxicity study with any of these compounds, should be also discussed.

Author Response

“The manuscript presented by Ciarlo et al. found a new molecule to increase NAD+ and a synergistic effect between two NAD+ modulators. The work is well structured and designed. Some minor issues should be fixed.

The authors use NRH abbreviator in the abstract without define it.”

We than the reviewer for his/her kind words on our work. NRH is now defined in the abstract

“In p.3 l.126 "Cells" should be “cells”"

Thank you very much for noticing this unintended mistake. This has now been corrected

“In section 2.5 also secondary antibodies should be listed”

Secondary antibodies are now also listed. We apologize for this omission.

“It should be desirable a graphic relating NR, NRH and NARH, if possible (similar to those in figs 1A and 2A). Moreover NRK1 is cited in the text, but is not present in any of these figures.”

We now include a summary in Figure 7, which relates the different paths and clarifies the role of NRK1 in them.

“In figure 3D (right panel, kidney) the effect of NR is similar to NR+NARH administered separately. In figure 5D (NRH and NADH) the is also different if the compounds are administered separately. It is important to comment and discuss these results.”

We thank the reviewer for this suggestion. A brief comment on these aspects can be found in Page 8, line 321 and Page 11, line 398

“The major limitation of the study is that, although these compounds are proposed as supplements, any toxicity assay is done. This limitation should be mentioned and discussed. If there are any previous toxicity study with any of these compounds, should be also discussed.”

This is an excellent point by the reviewer. We have added a few lines clarifying that there is a complete absence of toxicity studies on reduced NAD+ precursors (i.e. NRH and NARH) in lines 532-537.

Reviewer 2 Report

This is a very impressive paper and I do not see much necessity for improvement.

The major problem is the low number of experiments. I think this is only just acceptable since the results of various different experiments all point into the same direction. However, parametric statistics with an n between 3 and 5 remains problematic. This should be mentioned as a limitation. Note: for Figs 2B, C and 3B, C no n is given!

Some minor suggestions concerning the style:

The end of the introduction resembles a summary of the results. I´d rather suggest to better explain the questions aimed to be answered by the study and why these questions were asked.

On the other hand, some more explanations of the meaning of the results may be desired by the readers. Though, as mentioned, the physiological relevance of the described NAD+ precursors and their interactions remain to be elucidated, it would be interesting what this relevance could be and how the authors think (without getting too speculative) they could be translated to some clinical application one day.

Author Response

“This is a very impressive paper and I do not see much necessity for improvement.

The major problem is the low number of experiments. I think this is only just acceptable since the results of various different experiments all point into the same direction. However, parametric statistics with an n between 3 and 5 remains problematic. This should be mentioned as a limitation. Note: for Figs 2B, C and 3B, C no n is given!”

We agree with the reviewer on this potential limitation of the study, which is now explicitly mentioned in the discussion (Lines 542-543). We would also like to apologize for the unintended omission in the “n” number for the figures mentioned. This has now been corrected.

“Some minor suggestions concerning the style:

The end of the introduction resembles a summary of the results. I´d rather suggest to better explain the questions aimed to be answered by the study and why these questions were asked.”

Following the suggestion of the reviewer, we have now modified the last paragraph in the introduction, simply stating the rational of the study and the main questions addressed.

“On the other hand, some more explanations of the meaning of the results may be desired by the readers. Though, as mentioned, the physiological relevance of the described NAD+ precursors and their interactions remain to be elucidated, it would be interesting what this relevance could be and how the authors think (without getting too speculative) they could be translated to some clinical application one day.”

We have added a final paragraph in the discussion clarifying that the safety profiles of reduced NAD+ precursors (NRH or NARH) remain unknown and, therefore, these compounds not ready for clinical application. In addition, our study allows us to speculate that adding NARH might allow to improve NR based therapies. Indeed, when NR is administered alone it suffers from poor bioavailability and it is hardly detectable in circulation. In contrast, when administered together with NARH, it is robustly detected in circulation. Still, the mechanisms allowing for NR presence in circulation upon the NR/NARH co-treatment remain unknown.